# Nesting Preferences of *Osmia orientalis* (Hymenoptera: Megachilidae) in the Field and Its Potential as a Strawberry Pollinator in Greenhouses

**DOI:** 10.3390/insects16050473

**Published:** 2025-04-29

**Authors:** Ikuo Kandori, Yudai Ogata, Tomoyuki Yokoi

**Affiliations:** 1Laboratory of Entomology, Faculty of Agriculture, Kindai University, Naka-machi, Nara 631-8505, Japan; 2Faculty of Life and Environmental Sciences, University of Tsukuba, Tsukuba 305-8572, Japan; yokoi.tomoyuki.gp@u.tsukuba.ac.jp

**Keywords:** artificial plastic shells, *Euhadra amaliae*, per-visit pollination efficiency, snail shells, solitary bees

## Abstract

The mason bee, *Osmia orientalis*, native to Japan, is a promising crop pollinator. It exhibits the unique behavior of nesting in empty snail shells, although many aspects of its nesting behavior remain unknown. To investigate its nesting preferences, we placed the empty shells of four snail species in six different environments in the field. The bees showed a clear preference for the medium-sized shells of *Euhadra amaliae*. Regarding their surrounding environment, they preferred grasslands to bare ground and forest interiors. Next, to evaluate whether this bee can be used as a pollinator for strawberries as an alternative to the introduced western honeybee, *Apis mellifera*, we investigated the per-visit pollination efficiency of strawberry flowers in a greenhouse. The results suggested that *O. orientalis* has the same or greater pollination efficiency as *A. mellifera*. When *E. amaliae* shells and three-dimensionally printed plastic replicas were placed in a greenhouse, the bees nested equally frequently in *E. amaliae* shells and in several types of artificial shells. This result suggests that artificial shells could be used as alternative nesting materials when natural snail shells are scarce. Although many questions remain before practical application, our study supports the potential for using *O*. *orientalis* as a pollinator of greenhouse-grown strawberries.

## 1. Introduction

Insect pollinators are very important for the reproduction of wild plants and are used extensively in agricultural production [1,2]. The main insects used for pollinating crops are bees (Apoidea: Hymenoptera). The western honeybee, *Apis mellifera* L., plays a crucial role in crop pollination and is used worldwide [3]. In Japan, it is used for pollinating apple, watermelon, strawberry, and melon plants. However, honeybees tend to avoid certain crops, or pollinate them with low efficiency [4,5,6]. Additionally, the presence of the western honeybee as an introduced species in Japan could lead to competition with native wild bees for flower resources, which could potentially affect the biology and ecology of native bee species [7,8,9].

In recent years, the use of native bees as crop pollinators has attracted attention worldwide. *Osmia* bees (family Megachilidae) are all solitary and mostly univoltine. Some of these bees, such as *O*. *cornifrons* (Radoszkowski) in Japan, *O*. *lignaria* Say in the USA, and *O*. *cornuta* (Latreille) in Europe, have already been put to practical use [10]. Therefore, the ecology of many *Osmia* species and their characteristics related to crop pollination have been studied [11,12,13,14,15]. However, most of the species studied nest in holes in trees, or in bamboo or reed tubes. Snail shell nesting has been recorded in 52 osmiine bee species belonging to five genera, including *Osmia*, *Hoplitis*, and *Wainia* [16]. Little is known about these species’ nesting habits and no attempt has been made to utilize them agriculturally. Of the eight *Osmia* species found in Japan, only *O*. *orientalis* nests in empty snail shells in nature [12,17,18]. Like other *Osmia* bees, it is a potential pollinator of agricultural crops.

In this study, we conducted three experiments. In the first experiment, we investigated the nesting preferences of *O*. *orientalis* in the field, to collect basic information about its nesting habits. We examined which species of snail shells *O*. *orientalis* prefers for nesting, and what type of surrounding environment they favor while choosing a nesting site. *Apis mellifera* is commonly used as a pollinator in greenhouse strawberry (*Fragaria* × *ananassa* Duchesne ex Rozier) cultivation. In the second experiment, we evaluated the pollination effectiveness of *O*. *orientalis* on greenhouse strawberries by comparing its per-visit pollination efficiency with that of *A*. *mellifera*. In the third experiment, to develop an alternative nesting material to scarce natural empty snail shells, we created plastic shells that were three-dimensional (3D) replicas of empty *Euhadra amaliae* shells, the most preferred shells in the first experiment. These artificial shells were placed in greenhouses together with *E. amaliae* shells, and the bees’ nesting preferences were compared. Our results elucidate the little-known nesting habits of *Osmia* bees that nest in empty snail shells, and this is the first attempt to use them as agricultural pollinators.

## 2. Materials and Methods

### 2.1. Study Location and Study Species

All experiments were conducted in fields on the Nara Campus of Kindai University, Naka-machi, Nara, Japan (135.74° E, 34.67° N). The campus is a rural afforested environment surrounded by secondary forests with a high tree density. 

*Osmia orientalis* Benoist (Figure 1) is a univoltine bee that emerges as an adult in spring. While other species in the genus *Osmia* nest mainly in bamboo or reed tubes, this species has the unique habit of nesting only in empty snail shells [12,17,18]. Other *Osmia* bees found around the campus include *O*. *taurus* Smith and *O*. *jacoti* Cockerell, but *O*. *orientalis* is the most abundant species in the genus [19,20].

### 2.2. Experiment 1: Nesting Preferences for Natural Shells and Surrounding Environments in the Field

To identify the species of empty snail shells in which *O*. *orientalis* prefers to nest, we first prepared empty shells of four snail species (Figure 2). *Euhadra amaliae* (Kobelt) (Stylommatophora: Bradybaenidae) occurs in the Kansai region of Honshu. Its shell is of medium size, measuring 20–24 mm in height and 26–37 mm in diameter. The shell is pale yellowish-white, shiny, and somewhat thin, and the spiral spire is low and conical [21]. *Satsuma japonica* (Pfeiffer) (Stylommatophora: Camaenidae) is widely distributed in Honshu. Its shell is smaller, measuring 17 mm in height and 19 mm in diameter. The shell is thin and translucent, yellowish brown to dark brown, and the spiral spire is rounded and conical [21]. These two species were selected because they are the most common land snails in the study area. *Sinotaia quadrata histrica* (Gould) (Architaenioglossa: Viviparidae) is the only freshwater snail among the four species, and is found in rice fields, irrigation channels, and ponds throughout Japan. The shell is 35 mm high and 23 mm in diameter. *Helix lucorum* L. (Stylommatophora: Helicidae) is an edible snail that is sold commercially. It occurs from central Italy to Asia Minor. The shell height and diameter are both 50 mm. These two species were selected because they were readily available commercially.

Empty shells of *E*. *amaliae* and *S*. *japonica* were collected on campus, and only those without holes in the shells were used. The empty shells were classified as new, intermediate, or old, according to the freshness of the shell. We classified shells as “new” if they were very transparent and the inside could be seen. Shells that were whitish and opaque were classified as “old”. The remaining shells were classified as “intermediate”. *S*. *quadrata* and *H*. *lucorum* were purchased commercially and were all new at the time of purchase. For the experiment, 589 shells were prepared: 271 *E*. *amaliae*, 181 *S*. *japonica*, 96 *S*. *quadrata*, and 41 *H*. *lucorum*. The insides of all of the shells were washed with water and dried, and then the surface of each shell was numbered with black permanent marker for identification. These shells were divided into 32 groups, taking care to minimize bias in species and freshness among groups. To investigate the types of surrounding environments in which *O*. *orientalis* prefers to nest in the wild, empty shells were placed in 32 locations within the study area, divided into six environments: tall grass, short grass, adjacent to buildings, forest edge, inside forest, and bare ground (Table 1).

For each environment, empty shells were placed at four to six locations. At each location, a 1 × 1-m square area was enclosed with four 80-centimeter-long garden poles and hemp rope, and the shells were scattered on the ground surface. The shells were scattered in April 2009 and retrieved at the end of June to check for nests. This experiment was similar to our previous study [22]; however, to obtain more detailed results on the nesting preferences of this species, the scale of the experiment was expanded by increasing the number of shells placed from 247 to 589, the number of shell species from two to four by adding *S. quadrata* and *H. lucorum*, and the number of environments in which the shells were placed from five to six, by including a forest environment.

#### Data Analysis

We used generalized linear models (GLMs) with type III sums of squares to test for effects on *O*. *orientalis* nesting rates in empty shells with a binomial error distribution and logit link. The binary response variable was whether the bees nested in each shell or not (1/0). The species of shell (four species) and the environment (six environments) in which the shells were placed were treated as fixed effects. When the species of shell had a significant effect, post hoc multiple comparisons were performed among estimated marginal means of the four shell species using *t*-tests for pairwise contrasts and the sequential Bonferroni adjustment. There were almost no nests in shells other than *E*. *amaliae*, so we repeated the GLM with the above procedure using only the *E*. *amaliae* nesting data, and performed post hoc multiple comparisons among the six environments.

### 2.3. Experiment 2: Per-Visit Strawberry Flower Pollination Efficiency of Apis Mellifera and Osmia orientalis

As an index of the strawberry pollination ability of *A*. *mellifera* and *O*. *orientalis*, we investigated the per-visit pollination efficiency of the two bee species by examining the relationship between the number of visits to a flower and the percentage of fertilized achenes in a berry. Deformation of strawberry fruit is directly related to the percentage of fertilized achenes [23,24].

This experiment was conducted in a greenhouse (5.0 × 8.0 × 3.0 m high) for *A*. *mellifera* and in an outdoor cage (1.8 × 1.8 × 1.8 m) for *O*. *orientalis*. In advance, 30 planters (21.5 × 65 × 18 cm high) were prepared with three strawberry plants (variety ‘Toyonoka’) per planter. These planters were maintained in the outdoor cage, which was completely free of flower-visiting insects. A commercial honeybee colony of *A*. *mellifera* was placed in the greenhouse. For *O*. *orientalis*, 11 females and 10 males were released in the outdoor cage. When several intact strawberry flowers were blooming per planter, four planters with flowers were introduced into the greenhouse or cage. Then we continuously observed insect visits to intact flowers (Figure 3) and bagged the flowers with fine nylon mesh (mesh size 100 μm) when the number of visits to an individual flower reached a predetermined number from 1 to 10. We also prepared unvisited (zero-visit) flowers for each insect taxon, in which flowers were bagged before any insect visits. Subsequently, a paper label was affixed to the pedicel of each of these flowers, recording the dates of visits, the insect species that visited the flower, and the numbers of visits. When all of the flowers in a planter had been visited the planned numbers of times, we moved the planter back into an insect-free outdoor cage. At the beginning of the experiment, we attempted to prepare approximately the same number of flowers (six or seven) for each number of visits from 0 to 10 by each bee species. However, the experiment took longer than expected, and we could not prepare six flowers for some visits by the end of the strawberry flowering period.

Furthermore, the number of harvested berries decreased due to factors such as rotting and predation by ants or slugs. Finally, the number of harvested berries for each of the 0 to 10 visits by each bee species varied from one to seven. All berries were harvested at approximately 2 weeks after flowering, when fertilized and unfertilized achenes are easily distinguishable by the naked eye based on their size. We counted the total number of achenes and the number of fertilized achenes, and calculated the percentage of fertilized achenes for each berry. The experiment was conducted between 10:00 and 14:00 on sunny or cloudy days in April and May 2011.

#### Data Analysis

We estimated the per-visit pollination efficiency of each bee species using the formula:
*P* = 1 − (1 − *P*_0_) e^−*aN*^,where *P* is the percentage of fertilized achenes in a single berry, *N* is the number of insect visits to a single flower, and *P*_0_ is *P* at *n* = 0. In self-compatible, hermaphroditic plants such as strawberries, *P*_0_ > 0 because some achenes are fertilized by self-pollen grains without insect visits. Details of this model were provided in our previous study [25]. In this model, *a* is the per-visit pollination efficiency of the examined insect taxon. The rate of increase in the percentage of fertilized achenes is initially high, and then decreases gradually as the insect visits the flower more times. The percentage of fertilized achenes approaches 1 asymptotically. The model was fitted to data for each bee species to estimate *P*_0_ and *a*.

### 2.4. Experiment 3: Nesting Preference Between Artificial and Natural Shells in Greenhouses

As candidate nesting materials instead of empty *E*. *amaliae* shells, which were highly preferred for nesting (see the results of Experiment 1), we used an industrial CT scanner to scan fresh, undamaged empty shells of *E*. *amaliae* and obtained 3D data on the empty shells. Based on these data, we used a 3D printer to create four types of translucent plastic artificial shells that faithfully reproduced the internal structure of the empty *E*. *amaliae* shells (Figure 4 and Figure 5).

The nylon 12 shell was printed with a layer pitch of 0.1 mm and weighed 2.24 ± 0.04 g (mean ± SE, *n* = 10); the UV-cured acrylic resin (hereafter “acrylic”) shell was printed with a layer pitch of 0.016 mm and weighed 0.87 ± 0.01 g (*n* = 10); the polypropylene shell was printed with a layer pitch of 0.1 mm and weighed 2.41 ± 0.04 g (*n* = 20); and the acrylonitrile butadiene styrene (ABS) resin shell was printed with a layer pitch of 0.254 mm and weighed 2.06 ± 0.002 g (*n* = 20). For reference, new empty *E*. *amaliae* shells weighed 0.96 ± 0.09 g (*n* = 20). The creation of these artificial shells was outsourced to several sculpture-production companies.

This experiment was conducted in the spring in 2016 and 2017. In 2016, two greenhouses were used: the greenhouse that was used in Experiment 2 and another greenhouse of roughly the same size, located next to it. In 2017, only one greenhouse was used. To raise *O*. *orientalis* in greenhouses for the long term, we first placed about 30 strawberry planters, similar to those used in Experiment 2, in each greenhouse. Because strawberries alone do not provide enough food for *O*. *orientalis*, we introduced about 10 plants each per greenhouse of supplementary flowering plants, including *Brassica oleracea* var. *acephala* DC (Brassicales: Brassicaceae), *Borago officinalis* L. (Boraginales: Boraginaceae), *Vicia villosa* Roth (Fabales: Fabaceae), and *Vaccinium* spp. (Ericales: Ericaceae) (potted or planted in the ground). These plants were selected because *O. orientalis* often visited these plants in our preliminary observations in the field. New and old *E*. *amaliae* shells, as well as the four types of artificial shells, were placed in the greenhouse after identification numbers had been written on each type of shell with permanent marker. To reduce the risk of intrusion by ants or slugs, or that of people accidentally stepping on them, small terracotta flower pots (12 cm in diameter × 7 cm in height) were filled with soil and a shell was placed on the soil in each pot; these pots were placed randomly in the shade inside the greenhouse. Five shell types were used in 2016: new and old *E*. *amaliae* shells, and nylon 12, ABS, and polypropylene shells. Five shell types were used in 2017, the same types of empty shells as in 2016, except that polypropylene was replaced with acrylic. Ten shells of each type, totaling 50 shells, were placed in each greenhouse. Then, 10 females and 15–20 males of *O*. *orientalis*, newly hatched within 1 week from nesting shells that had been stored in advance, were released into each greenhouse. The number of active adult females (number of nesting individuals) in each greenhouse was monitored approximately once a week. Additional females were released as necessary so that the number of active and additional released individuals totaled 10. The release of *O*. *orientalis* into the greenhouses began in late March in both years, and nesting activity ended in late May in 2016 and early June in 2017, when all of the shells were collected. Then, each shell was checked for the presence or absence of nests, and the nesting rate for each type of shell was calculated. A total of 31 females were released in two greenhouses in 2016 and 22 were released in one greenhouse in 2017. The lifetime number (mean ± standard deviation) of shells used by each female once it began nesting in greenhouses was 2.30 ± 1.61 (N = 23) in 2016 and 2.19 ± 1.11 (N = 16) in 2017.

#### Data Analysis

We used GLMs with type III sums of squares to test for effects on the nesting rates of *O*. *orientalis* in empty shells with a binomial error distribution and a logit link. Because the types of empty shell placed differed slightly, the statistics for 2016 and 2017 were processed separately. In addition, the data for 2016 were pooled as the combined total from the two greenhouses. Whether the bees nested in each shell or not (1/0) was used as the binary response variable. The types of empty shell (five types in each year) were treated as fixed effects. When the types of empty shell had a significant effect, post hoc multiple comparisons were performed among estimated marginal means of the five types of empty shell using *t*-tests for pairwise contrasts and sequential Bonferroni adjustment. IBM SPSS statistics 28 was used for all statistical analyses [26].

## 3. Results

### 3.1. Nesting Preferences for Natural Shells and Surrounding Environments in the Field

Some of the shells placed in the field could not be retrieved because they were lost, destroyed, or had ants nesting in them. Some shells were infested with ants, but they were all retrieved. If a shell still contained any pollen collected by the female bees, it was considered to have been used as a nest. Of the shells that could be retrieved, the percentage of shells in which *O*. *orientalis* had nested (nesting rate) was 8.7% (47/540). GLM analysis of factors influencing the nesting rate in empty shells showed that both shell species (likelihood ratio χ^2^ = 41.784, df = 3, *p* < 0.001) and surrounding environment (likelihood ratio χ^2^ = 33.699, df = 5, *p* < 0.001) had significant effects.

The nesting rate in empty shells was significantly higher for *E*. *amaliae* (16.1%) than for *S*. *quadrata* (3.7%), *S*. *japonica* (1.3%), or *H*. *lucorum* (0%) (post hoc multiple comparisons: *p* < 0.05; Figure 6).

Regarding the surrounding environment, the nesting rates in *E*. *amaliae* shells were significantly higher in tall grass (28.6%), in short grass (23.5%), adjacent to buildings (15.2%), and at the forest edge (18.0%) than in forest (0%) or on bare ground (0%) (post hoc multiple comparisons: *p* < 0.05; Figure 7). The nesting rates in *E*. *amaliae* shells tended to be particularly high in both tall and short grass (both over 20%).

### 3.2. Per-Visit Strawberry Flower Pollination Efficiency of A. mellifera and O. orientalis

Based on survey data on the relationship between the number of visits to strawberry flowers and the percentage of fertilized achenes in each berry (Figure 8), we performed a nonlinear regression analysis. The estimated regression equations were as follows:

*O*. *orientalis*: *P* = 1 − (1 − 0.306) e^−0.114N^ (*n* = 47) and

*A*. *mellifera*: *P* = 1 − (1 − 0.340) e^−0.091N^ (*n* = 55)

The pollination efficiency per flower visit, *a*, was estimated to be 0.114 for *O*. *orientalis* and 0.091 for *A*. *mellifera* (Figure 8), indicating that the pollination efficiency of *O*. *orientalis* per flower visit was equal to or greater than that of *A*. *mellifera*.

### 3.3. Nesting Preference Between Artificial and Natural Shells in Greenhouses

The nesting rates of *O*. *orientalis* in the various empty shells placed in the greenhouses differed significantly depending on the type of shell in both years (GLM; 2016: likelihood ratio χ^2^ = 70.073, df = 4, *p* < 0.001; 2017: likelihood ratio χ^2^ = 34.220, df = 4, *p* < 0.001). Of the five types of shells placed in 2016, the nesting rates in nylon 12, polypropylene, and new *E*. *amaliae* shells were similarly high, and were significantly higher than those in old *E*. *amaliae* and ABS resin (post hoc multiple comparisons *p* < 0.05; Figure 9). Similarly, of the five types of shells placed in 2017, the nesting rates in nylon 12, acrylic, and new *E*. *amaliae* were similarly high, and were significantly higher than those in old *E*. *amaliae* and ABS resin (post hoc multiple comparisons *p* < 0.05; Figure 9), except that there were no significant differences between old *E*. *amaliae* shells and acrylic shells.

## 4. Discussion

### 4.1. Nesting Preferences for Natural Shells and Surrounding Environments in the Field

#### 4.1.1. Nesting Preferences for Natural Shells

As reported in our previous study [22], wild *O*. *orientalis* nested in empty snail shells artificially placed in the field. Nesting rates in *E. amaliae* shells did not differ significantly between this study (16.1%, 42/261) and our previous study (24.1%, 32/133) (Fisher’s exact test: *p* = 0.076). Among the shell species used, the nesting rate was highest by far in *E*. *amaliae*, indicating that *O*. *orientalis* has a strong preference for these shells (Figure 6). *Osmia orientalis* showed little interest in *S*. *japonica* or *S*. *quadrata* shells. These shells are smaller than those of *E*. *amaliae* and have a smaller internal volume. When the internal volume is as small as that of *S*. *japonica*, all *O*. *orientalis* offspring in the nest become males [22], possibly because it is impossible to construct cells of female offspring, which are larger than those of male offspring. This may be one reason why *O*. *orientalis* does not prefer small shells such as *S*. *japonica* or *S*. *quadrata*. These results are consistent with those of our previous study, in that *O*. *orientalis* greatly preferred *E. amaliae* shells over *S*. *japonica* shells [22]. Furthermore, *O*. *orientalis* did not nest in *H*. *lucorum* shells (Figure 6), implying that they found these shells to be too large. Like several other osmiine bees that have been reported to manipulate and translocate nest shells [16], *O*. *orientalis* often used their hind legs to push shells around, or to position them upright with their opening facing downwards, possibly to prevent rain from entering (Kandori personal observations). Shells that are too large may not be chosen because they are too heavy for the bees to manipulate (for reference, empty *H*. *lucorum* shells weigh 4.97 ± 0.45 g [*n* = 12], which is approximately five times heavier than *E*. *amaliae* shells). However, different species of *Osmia* bees prefer different shell sizes, and *O. aurulenta* (Panzer) in Europe prefers the empty shells of large *Helix* snails [27]. Such *Osmia* species may nest without moving the shell. Numerous shell-nesting species do not change the position or orientation of the shells [16].

#### 4.1.2. Nesting Preferences for the Surrounding Environment

In terms of the environment in which the shells were placed, *O*. *orientalis* nested in tall grass, short grass, adjacent to buildings, and at the forest edge, but did not nest inside forests or on bare ground (Figure 7). These results were consistent with those of our previous study, in that nesting rates tended to be particularly high in shells placed in tall and short grass [22]. *Osmia orientalis* prefers to visit grass flowers rather than tree flowers [12,19,20], which may be related to its preference for grasslands as a surrounding environment for nesting. The nesting rate within a forest established as a new environment in this study was zero. The bees did not choose dark environments such as forest interiors or environments lacking a place to hide their nest shells, such as bare ground, preferring environments that are moderately covered by vegetation. As in our previous study [22], we found that they used artificial environments, such as the eaves of buildings, as long as the shells were adequately hidden. Their ability to use artificial environments may be related to the greenhouse-nesting ability observed in Experiment 3.

### 4.2. Per-Visit Strawberry Flower Pollination Efficiency of A. Mellifera and O. orientalis

The per-visit strawberry flower pollination efficiency of *O*. *orientalis* was equal to or greater than that of *A*. *mellifera* (Figure 8). Several studies have shown that the pollination ability of *Osmia* bees (percentage of fruits set after single visits, percentage of visits in which the stigmas were contacted, number of flowers visited by one individual per unit of time, interplant movement frequency, and so forth) is equal to or greater than that of *A*. *mellifera* [28,29,30,31,32]. In future studies, it will be necessary to compare the efficacy or importance of *O*. *orientalis* and *A*. *mellifera* as strawberry pollinators, which can be calculated as the product of per-visit pollination efficiency and visit frequency [25,33,34,35,36].

### 4.3. Nesting Preference Between Artificial and Natural Shells in Greenhouses

In this study, *O*. *orientalis* in a greenhouse nested both in the shells of *E*. *amaliae* and in artificial plastic shells made from nylon 12, acrylic, and polypropylene (Figure 9). Furthermore, the artificial shells were chosen as often as empty *E*. *amaliae* shells, implying that they can be used as nesting materials for *O*. *orientalis*. As a next step, it is necessary to compare reproduction-rate indicators, such as the number of cells per shell, the next-generation emergence rate (survival rate to adulthood), and the next-generation sex ratio, between natural and artificial shells.

Of the four types of artificial shells tested in this study, only ABS resin had a low nesting rate (Figure 9). This was probably because the layer pitch of the 3D printer was rough (or thick), causing the inner and outer surfaces of the shell to be subtly wavy, which made it impossible to reproduce the smooth texture of snail shells.

The use of petroleum-based plastics raises issues such as resource depletion and global warming. Additionally, discarded persistent plastics contribute to marine pollution, negatively impacting marine ecosystems [37,38,39]. All four types of plastics used as raw materials for artificial shells in this study were petroleum-based, persistent plastics. These problems could be avoided in future studies by constructing artificial shells from biodegradable biomass plastic.

## 5. Conclusions

Our results demonstrate the possibility of using *O*. *orientalis* to pollinate greenhouse-grown strawberries. First, wild *O*. *orientalis* nested in empty snail shells that were placed outdoors. In this way, the next generation of adult *O*. *orientalis* bees can be obtained and released into greenhouses as agricultural pollinators. Second, *O*. *orientalis* visited greenhouse strawberry flowers, and per-visit pollination efficiency was equal to or better than that of *A*. *mellifera*. This indicates that *O*. *orientalis* has a high pollination ability, like other species of *Osmia* bees, and that it could be used as an agricultural pollinator, even in closed greenhouses. Third, *O*. *orientalis* frequently nested in both natural and artificial snail shells that were placed inside the greenhouse, indicating that *O*. *orientalis* can breed in greenhouses and use artificial shells as nesting materials. Although many challenges remain before practical use, our findings increase the probabilities of *O. orientalis* as a pollinator for greenhouse-grown strawberries.

## Figures and Tables

**Figure 1 insects-16-00473-f001:**
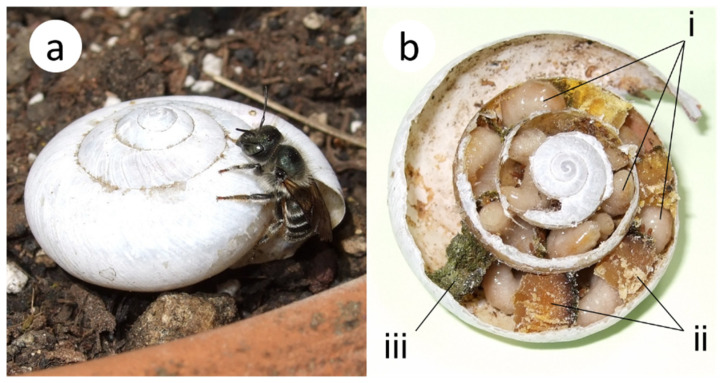
*Osmia orientalis* nesting in an empty shell of the snail *Euhadra amaliae*. (**a**) Female adult on a shell. (**b**) Larvae growing inside a shell. (**i**) Larvae. (**ii**) Pollen-nectar provision. (**iii**) Plug used to close the nest.

**Figure 2 insects-16-00473-f002:**
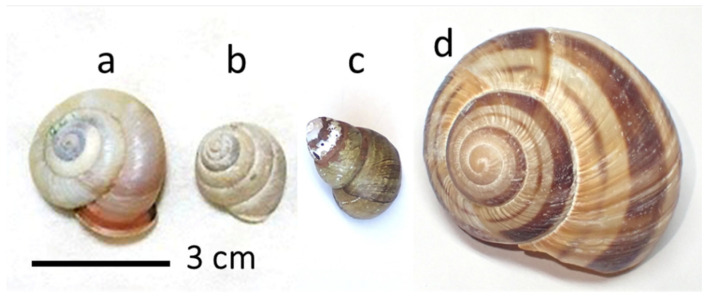
Empty shells of four snail species placed in the field: (**a**) *Euhadra amaliae*, (**b**) *Satsuma japonica*, (**c**) *Sinotaia quadrata histrica*, and (**d**) *Helix lucorum*.

**Figure 3 insects-16-00473-f003:**
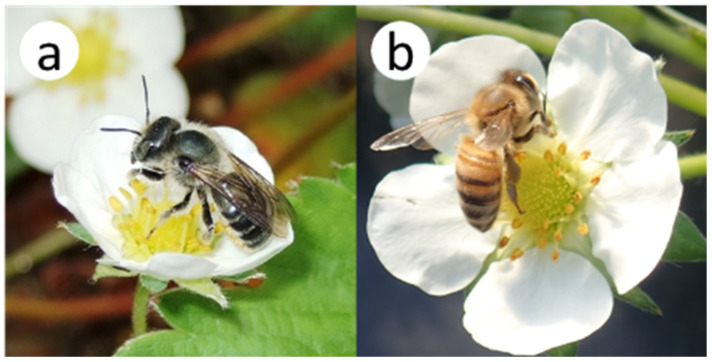
Two bee species visiting strawberry flowers: (**a**) *Osmia orientalis* and (**b**) *Apis mellifera*.

**Figure 4 insects-16-00473-f004:**
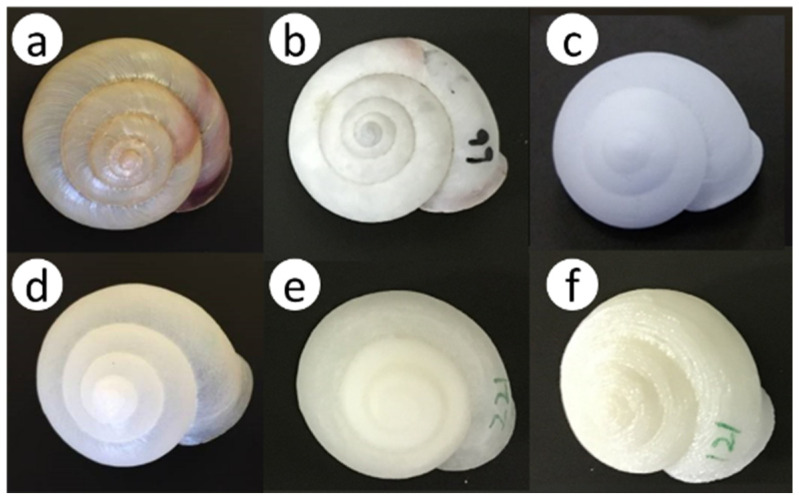
Two types of natural shells and four types of plastic artificial shells placed in greenhouses: (**a**) new *Euhadra amaliae* shell; (**b**) old *E*. *amaliae* shell; and artificial shells made of (**c**) nylon 12, (**d**) acrylic, (**e**) polypropylene (PP), and (**f**) acrylonitrile butadiene styrene (ABS) resin.

**Figure 5 insects-16-00473-f005:**
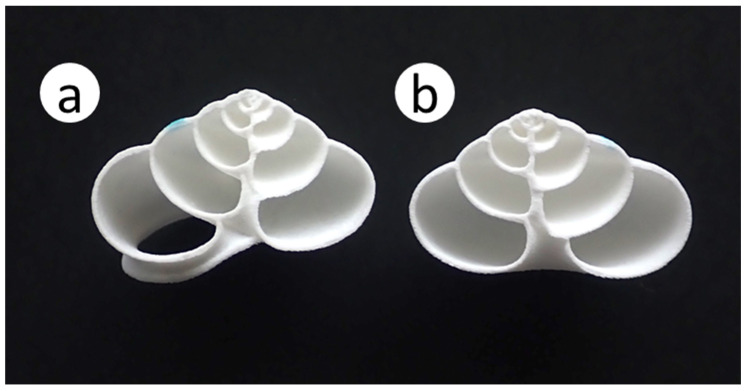
The inside of an artificial shell made of nylon 12, cut in half along the center line. The (**a**) left and (**b**) right halves of the shell.

**Figure 6 insects-16-00473-f006:**
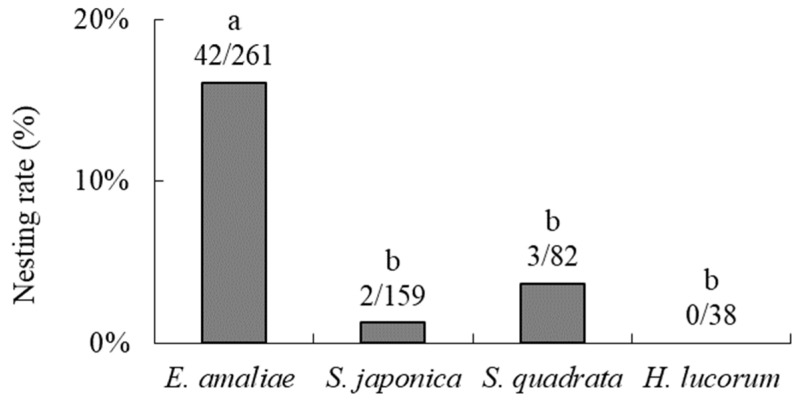
Nesting rates of *Osmia orientalis* in the shells of four snail species in the field: *Euhadra amaliae*, *Satsuma japonica*, *Sinotaia quadrata histrica*, and *Helix lucorum*. Numbers indicate the numbers of recovered shells that contained nests and the total numbers of shells recovered. Different letters above the bars indicate significant differences among species (post hoc Bonferroni *p* < 0.05).

**Figure 7 insects-16-00473-f007:**
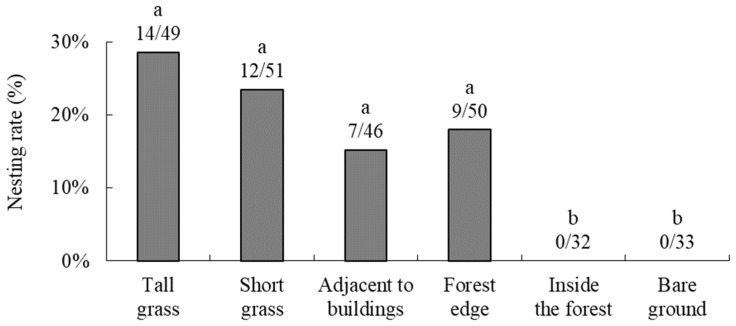
Nesting rates of *Osmia orientalis* in *Euhadra amaliae* shells that were placed in six environments in the field. Numbers are the numbers of recovered shells that contained nests and the total numbers of shells recovered. Different letters above the bars indicate significant differences among environments (post hoc Bonferroni *p* < 0.05).

**Figure 8 insects-16-00473-f008:**
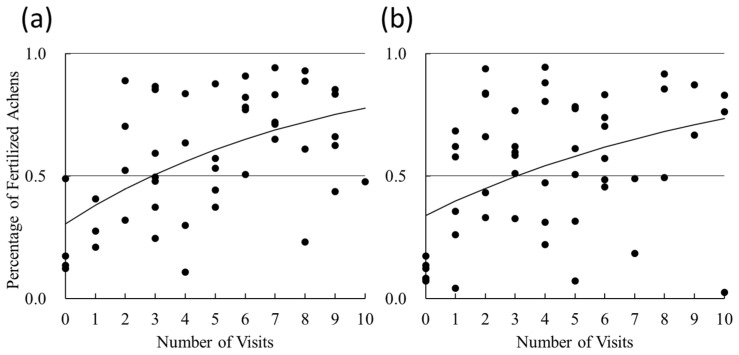
Relationship between the number of insect visits to a strawberry flower and the percentage of fertilized achenes in a berry for (**a**) *Osmia orientalis* and (**b**) *Apis mellifera*. Nonlinear regression lines are shown (see Section 3 for details).

**Figure 9 insects-16-00473-f009:**
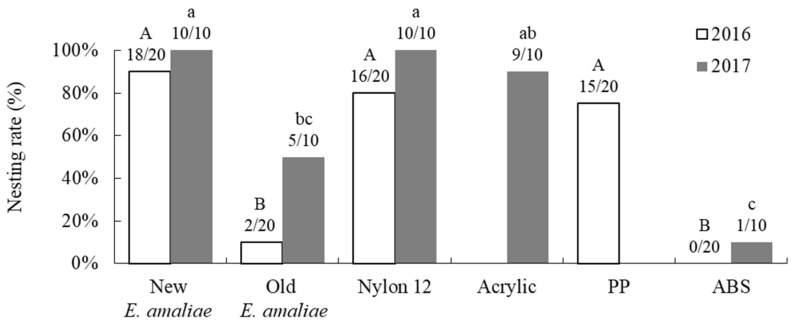
Nesting rates of *Osmia orientalis* in natural new and old *Euhadra amaliae* shells, and in four types of artificial shells (nylon 12, acrylic, PP, and ABS resin), in a greenhouse. The different letters above the bars indicate significant differences among shell types within each year (post hoc Bonferroni *p* < 0.05).

**Table 1 insects-16-00473-t001:** The six environments in which empty snail shells were placed.

Environment (Number of Locations)	Environmental Characteristics
Tall grass (*n* = 6)	Herbaceous plants over 1 m in height (e.g., ferns, goldenrod *Solidago altissima*, and silver grass *Miscanthus sinensis*) grew densely. Direct sunlight was rare.
Short grass (*n* = 6)	Herbaceous plants, approximately 30 cm in height, grew somewhat densely. Frequent direct sunlight.
Adjacent to buildings (*n* = 5)	Shade or semi-shade under the eaves or rim of buildings, such as work sheds and school buildings. There was almost no rain and no viable grass.
Forest edge (*n* = 6)	Half-shade at the edge of a forest. Fallen leaves were piled up, and viable grass was sparse.
Within a forest (*n* = 5)	Approximately 10 m inside a broad-leaved forest dominated by *Quercus serrata* and *Q*. *glauca*. The area was shaded all day with almost no viable grass.
Bare ground (*n* = 4)	A 2-m-radius circle with bare soil and no growing plants. We weeded the area, which originally had relatively little grass. The area received direct sunlight all day.

## Data Availability

The original contributions presented in this study are included in the article. Further inquiries can be directed to the corresponding author.

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
