# Peer review of "Nesting Preferences of Osmia orientalis (Hymenoptera: Megachilidae) in the Field and Its Potential as a Strawberry Pollinator in Greenhouses"

_insects, 2025, doi:10.3390/insects16050473_

Round 1

Reviewer 1 Report

Comments and Suggestions for Authors

This is a straight forward set of experiments examining the nesting preferences of a native solitary bee in Japan, and its potential as a pollinator of greenhouse grown strawberries. This paper adds some basic knowledge to our understanding of potentially useful pollinator with practical implications for strawberry pollination specifically in Japan.

Experimental design and analysis are mostly appropriate. There is a lack of replication on habitat – this variable is technically pseudo-replicated. Multiple locations of the same habitat type would need to be used to definitively say that certain habitats are preferred over others. This should be acknowledged in the interpretation.

Line 28: Implies that shell weight was an analyzed variable. Suggest removing the part about weight.

Line 50: “Osmia bees” – include family and basic information on what an “Osmia bee” is.

Line 59: Sentence beginning “in this study…”  The descriptions of the experiments in the introduction here, and in subsequent paragraph are not necessary as it creates unnecessary repetition. This is a short, straightforward paper and the experiments will be described in the methods section immediately following. I suggest removing all detailed explanations of the 3 experiments, shortening it to 3 sentences that simply summarize what will be done.

This includes lines 59-64, 71-74 and lines 77-83.

Line 91-93: Relatively high preference as compared to what? This might be worth expanding in the introduction rather than here.

Line 169: What was the goal sample size for each visit number? What level of replication was actually reached?

Line 219: Why these plants?

Line 307: This seems unnecessary as a separate two line paragraph. Start with 4.1.1 and add to the beginning

Line 340: Needs citations – there are many studies that take per-visit pollination efficiency and combine with in field visit rates to examine efficacy of different species

Line 343: What are these “related factors”?

Line 361: I would call this ‘per visit’ pollination efficiency, since you don’t know overall efficiency yet

Author Response

Responses to Reviewer 1

[Manuscript ID: insects-3545977]

Title:

"Nesting preferences of Osmia orientalis (Hymenoptera: Megachilidae) in the field and its potential as a strawberry pollinator in greenhouses"

Dear Reviewer 1,

Thank you very much for your effort of giving us valuable comments. Below is the specific response to Reviewer 1.

The major changes include a major rewrite of the Simple Summary, shortening and simplifying the latter half of the Introduction, adding many references to the Discussion, and changing Supplementary Figures S1 and S2 to Figures 3 and 5.

We indicate how we responded to the reviewer's comments as follows:

--------------------------------------------------------------------------------

>> Reviewer's comments<<

REPLY number

Our responses (line number where the change occurred in the 'manuscript with track changes')

--------------------------------------------------------------------------------

>> Line 28: Implies that shell weight was an analyzed variable. Suggest removing the part about weight. <<

REPLY 1

We removed ‘lightweight’ according to your suggestion. (L. 22)

>> Line 50: “Osmia bees” – include family and basic information on what an “Osmia bee” is <<

REPLY2

We included family name ‘Megachilidae’ and basic information that ‘they are all solitary and mostly univoltine’ as you suggested. (L. 79)

>> Line 59: Sentence beginning “in this study…”  The descriptions of the experiments in the introduction here, and in subsequent paragraph are not necessary as it creates unnecessary repetition. This is a short, straightforward paper and the experiments will be described in the methods section immediately following. I suggest removing all detailed explanations of the 3 experiments, shortening it to 3 sentences that simply summarize what will be done. <<

>> This includes lines 59-64, 71-74 and lines 77-83.<<

REPLY 3

By following your suggestion, we removed about half of the former sentences. Unfortunately, we were unable to fit the content of each experiment into one sentence, but we managed to fit it into two sentences each. (L. 92-121).

>> Line 91-93: Relatively high preference as compared to what? This might be worth expanding in the introduction rather than here.<<

REPLY 4

We intended to write O. orientalis has relatively high preference for strawberry flowers as compared to other common flowers. However, since it is self-evident from the results of Experiment 2 that O. orientalis frequently visits strawberry flowers, this sentence has been deleted from the manuscript.(L. 133-134)

>> Line 169: What was the goal sample size for each visit number? What level of replication was actually reached? <<

REPLY 5

At the beginning of the experiment, we set the goal sample size for each visit number as six or seven. However, the experiment took longer than expected, and we could not prepare six flowers for some visits by the end of the strawberry flowering period. The actually reached level was varied from one to seven for each of the 0 to 10 visits by each bee species. We added this information to the manuscript. (L.231-245).

>> Line 219: Why these plants? <<

REPLY 6

These plants were selected because O. orientalis often visited these plants in our preliminary observations in the field. We added this sentence to the manuscript. (L.302-304).

>> Line 307: This seems unnecessary as a separate two line paragraph. Start with 4.1.1 and add to the beginning<<

REPLY 7

Following your comment, we moved this sentence to the beginning of 4.1.1. (L.414-418).

>> Line 340: Needs citations – there are many studies that take per-visit pollination efficiency and combine with in field visit rates to examine efficacy of different species <<

REPLY 8

In the future experiments, we limited our focus to examining the number of flower visits per unit time and calculating efficacy or pollinator importance. Then we included several references on this topic. (L.468-474).

>> Line 343: What are these “related factors”? <<

REPLY 9

Regarding REPLY 8, we reduced the experiments to be conducted in the future. Therefore, we have removed the sentence containing "related factors." (L. 473).

>> Line 361: I would call this ‘per visit’ pollination efficiency, since you don’t know overall efficiency yet <<

REPLY 10

Following your comment, we changed it to ‘per visit’ pollination efficiency. (L.500).

Reviewer 2 Report

Comments and Suggestions for Authors

This is a fascinating paper which explores the utilisation of a solitary bee for crop pollination, which required investigation into the nesting ecology and preferences. The experimental design seems sound, and it seems like a potentially effective and novel technique. However, it’s unclear how many individual bees survived in a closed environment, how many would be required to pollinate X crops, and associated quantifications regarding the resources this species requires. There is also a lack of engaging with the literature across introduction and discussion. 

Abstract / summary:

  • I’m not certain of the differences between the simple summary and abstract other than word count, I think the simple summary needs some more work.
  • I think even with the abstract there should be at least a sentence on the context of the study. Indeed it isn’t clear in either the simple summary or the abstract what the problem is, and why the study is important.
  • It would be interesting to know in the abstract what the other types of shell were, similar to the habitats.
  • In the abstract it states ‘was estimated to be’ - some clarity required here. Why not just our results indicated the pollinators efficiency was equal or greater.
  • Presumably the artificial shells could be produced for crop pollination purposes given naturally occurring shells would be difficult to come by (and probably not a good idea to collect for this purpose), this could be made clearer in the abstract.
  • The take home is these findings indicate the potential of using this species in this context, as there are a lot of unanswered questions. I think this needs to be made clearer in the final sentence of both summary and abstract.

Line 56 Reconsider using the word ‘exploit’. Utilise?

Line 60+ It would be useful to know the decision making here regarding selection shells, just a comment that this was based on what’s commonly found.

Line 235+ it would be useful to know how many had to be released, presumably due to death? If this was monitored weekly presumably these data are available. Do we know anything about how many shells one female nests in, presumably it’s possible it was just one female.

Line 307 reconsider having a section with one sentence, integrate this into the paragraph, and how does the rate compare to the previous paper, <16% success rate as well? Or start with an overview of the major findings, the greenhouse % success rate of shells is quite significant.

Line 305+ - there’s no real attempt at engaging with the literature in this section, comparing findings with those published across all sections. This is more summarising the results again, albeit also summarising a clear and well written application.

Author Response

Responses to Reviewer 2

[Manuscript ID: insects-3545977]

Title:

"Nesting preferences of Osmia orientalis (Hymenoptera: Megachilidae) in the field and its potential as a strawberry pollinator in greenhouses"

Dear Reviewer 2,

Thank you very much for your effort of giving us valuable comments. Below is the specific response to Reviewer 2.

The major changes include a major rewrite of the Simple Summary, shortening and simplifying the latter half of the Introduction, adding many references to the Discussion, and changing Supplementary Figures S1 and S2 to Figures 3 and 5.

We indicate how we responded to the reviewer's comments as follows:

--------------------------------------------------------------------------------

>> Reviewer's comments<<

REPLY number

Our responses (line number where the change occurred in the 'Revision manuscript with track changes')

--------------------------------------------------------------------------------

>> Abstract / summary:

I’m not certain of the differences between the simple summary and abstract other than word count, I think the simple summary needs some more work. <<

REPLY 1

We have made significant changes to Simple Summary. (L. 15-37)

>> I think even with the abstract there should be at least a sentence on the context of the study. Indeed it isn’t clear in either the simple summary or the abstract what the problem is, and why the study is important. <<

REPLY2

In both the simple summary and the abstract, we wrote at the beginning what the problem is. (L. 17-18, 39-40)

>> It would be interesting to know in the abstract what the other types of shell were, similar to the habitats.<<

REPLY 3

We added the information about empty shells of four snail species to the abstract (L. 46-47). But according to six different environments, we are sorry, but we couldn't add more detail information due to word limit.

>> In the abstract it states ‘was estimated to be’ - some clarity required here. Why not just our results indicated the pollinators efficiency was equal or greater.<<

REPLY 4

Following your suggestion, we changed the sentence in both the simple summary and the abstract. (L. 27-30, 52-53)

>> Presumably the artificial shells could be produced for crop pollination purposes given naturally occurring shells would be difficult to come by (and probably not a good idea to collect for this purpose), this could be made clearer in the abstract.<<

REPLY 5

We added the reason for developing artificial shells, that is, natural shells are scarce. (L. 54). We also added a relevant content to the simple summary. (L. 34-35)

>> The take home is these findings indicate the potential of using this species in this context, as there are a lot of unanswered questions. I think this needs to be made clearer in the final sentence of both summary and abstract.<<

REPLY 6

As you suggested, we added the sentences like ‘there are a lot of unanswered questions’ in the final sentence of both summary and abstract. (L.35-36, 58).

>> Line 56 Reconsider using the word ‘exploit’. Utilise?<<

REPLY 7

Following your comment, we changed the word ‘exploit’ to ‘utilize’. (L.89).

>> Line 60+ It would be useful to know the decision making here regarding selection shells, just a comment that this was based on what’s commonly found.<<

REPLY 8

Euhadra amaliae and Satsuma japonica were selected because they are the most common land snails in the study area. Sinotaia quadrata and Helix lucorum were selected because they were readily available commercially. These sentences were added to the manuscript (L.155, 161-162).

>> Line 235+ it would be useful to know how many had to be released, presumably due to death? If this was monitored weekly presumably these data are available. Do we know anything about how many shells one female nests in, presumably it’s possible it was just one female. <<

REPLY 9

Following your comment, we added the information about how many individual bees were released and how many shells one female nests in the greenhouse. (L.322-325).

>> Line 307 reconsider having a section with one sentence, integrate this into the paragraph, and how does the rate compare to the previous paper, <16% success rate as well? Or start with an overview of the major findings, the greenhouse % success rate of shells is quite significant. <<

REPLY 10

Following your comment, we moved this sentence to the beginning of 4.1.1. We also added the sentence that there was no significant difference in the nesting rates in E. amaliae shells between this study and our previous study. (L.414-420).

>> Line 305+ - there’s no real attempt at engaging with the literature in this section, comparing findings with those published across all sections. This is more summarising the results again, albeit also summarising a clear and well written application. <<

REPLY 11

Following your comment, we added several literatures here and there and compared our results with some of those literatures. (L.438-442, 449-451, 468-474, 488-493).

Reviewer 3 Report

Comments and Suggestions for Authors

The authors have investigated the nesting preferences and pollination efficiency of Osmia orientalis to assess its potential as a strawberry pollinator under greenhouse conditions. The experiments showed that the bee prefers medium-sized, lightweight Euhadra amaliae snail shells for nesting and is as effective as Apis mellifera in providing pollination services to strawberry flowers. Artificial plastic shells showed similar nesting rates, which suggests their potential as viable alternatives for managed pollination. My comments are:

L27: I would suggest putting the scientific name here.

L28: Abbreviate the genus name here, as the full name came in L27.

L45: Replace: However, honeybees do not prefer certain crops, or they pollinate them with low efficiency.

L46-48: Replace: In addition, while the Western honeybee plays a crucial role in pollination, its presence as an introduced species in Japan could lead to competition with native wild bees for floral resources, which could potentially affect the biology and ecology of native bee species. (Find some references that could also highlight the positive role of A. mellifera. We need to be just while highlighting both aspects of a species.)

L49-51: Merge it with the following paragraph, as you are building the same case in the following sentences.

L54-55: I prefer to use three and one, instead of 3 and 1, respectively.

L59: Replace: In the first experiment…

L60-62: We examined which species of snail shells O. orientalis prefer for nesting and the type of surrounding environment they prefer while choosing a nesting site.

L69-70 and overall manuscript: Use complete scientific name, authority, order and family, when you are using the species names for the first time.

L68-71: Maintain clarity: Recently in Japan, native pollinators like ABC and XYZ have been used to replace or supplement Apis mellifera in greenhouse strawberry cultivation. However, these native pollinators have not yet been consistently adopted across a large scale.

L71, 77 and 80: I suggest using the first experiment, second experiment, and third experiment.

Figure 1b: I would label the figure 1b for educating the early career researchers. And describe it a little more in the caption.

Materials and methods are well-described.

L253: Please clarity whether it was ant nesting or ant infestation. Both are different.

L265: Remove see results for details. It is understood.

L276 and 304: Same comment as for L265.

L306-308: Please expand the discussion around your previous study. Highlight the deficiencies/future directions and how did you expand further in the current study. This will educate the early researchers in the scientific process.

L311: What could be the reasons for higher preference for this shell? Is it the diameter or some architectural features? Think of the body diameter, larval size and other factors here.

L327: How are these consistent with earlier study? Do explain when you say this.

L344: Please also discuss the possibility of using other biomaterials, as these plastic-based shells will add the pollution factor. So, if we can construct some shells using biomaterials and 3D printers, then what could be the possibilities and what could be the alternate materials which could be used. That may also provide the base for future studies.

Add one paragraph of future directions.

L359, 362, 366: Please delete experiment 1, 2, and 3 in brackets. It is understood and clear.

Please move supplementary figures as the main figures. These are needed in the main text.

Overall, well done. I appreciate the hard work.

Author Response

Responses to Reviewer 3

[Manuscript ID: insects-3545977]

Title:

"Nesting preferences of Osmia orientalis (Hymenoptera: Megachilidae) in the field and its potential as a strawberry pollinator in greenhouses"

Dear Reviewer 3,

Thank you very much for your effort of giving us valuable comments. Below is the specific response to Reviewer 3.

The major changes include a major rewrite of the Simple Summary, shortening and simplifying the latter half of the Introduction, adding many references to the Discussion, and changing Supplementary Figures S1 and S2 to Figures 3 and 5.

We indicate how we responded to the reviewer's comments as follows:

--------------------------------------------------------------------------------

>> Reviewer's comments<<

REPLY number

Our responses (line number where the change occurred in the 'Revision manuscript with track changes')

--------------------------------------------------------------------------------

>> L27: I would suggest putting the scientific name here. <<

REPLY 1

We added the scientific names of the four snail species, according to your suggestion. (L. 46-47)

>> L28: Abbreviate the genus name here, as the full name came in L27. <<

REPLY2

We abbreviated the genus name, according to your suggestion. (L. 48)

>> L45: Replace: However, honeybees do not prefer certain crops, or they pollinate them with low efficiency.<<

REPLY 3

“However, honeybees tend to avoid certain crops, or pollinate them with low efficiency.” The above minor correction was made during English proofreading, so we replaced it with this sentence. (L. 71-73).

>> L46-48: Replace: In addition, while the Western honeybee plays a crucial role in pollination, its presence as an introduced species in Japan could lead to competition with native wild bees for floral resources, which could potentially affect the biology and ecology of native bee species. (Find some references that could also highlight the positive role of A. mellifera. We need to be just while highlighting both aspects of a species.)<<

REPLY 4

We added the statement "Western honeybees play an important role in crop pollination" to the above sentence, along with a new reference. (L.68-69)

We slightly modified the latter half of the sentence you suggested and replaced it with the previous sentence as follows: “Additionally, the presence of the western honeybee as an introduced species in Japan could lead to competition with native wild bees for flower resources, which could potentially affect the biology and ecology of native bee species.” (L.73-77)

>> L49-51: Merge it with the following paragraph, as you are building the same case in the following sentences.<<

REPLY 5

We merge it with the following paragraph, as you suggested. (L. 82-83)

>> L54-55: I prefer to use three and one, instead of 3 and 1, respectively. <<

REPLY 6

In the process of obtaining new literature information according to the instructions of other referees, the previous texts and relevant parts were replaced with the new texts and literature. (L.85-88).

>> L59: Replace: In the first experiment… <<

REPLY 7

Following your comment, we replaced with ‘In the first experiment’ (L.93), ‘In the second experiment’ (L.110), and ‘In the third experiment’, (L.117).

>> L60-62: We examined which species of snail shells O. orientalis prefer for nesting and the type of surrounding environment they prefer while choosing a nesting site.<<

REPLY 8

Following your comment, we made some slight edits to the text you provided and replaced the relevant text in the manuscript. (L.95-98).

>> L69-70 and overall manuscript: Use complete scientific name, authority, order and family, when you are using the species names for the first time.<<

REPLY 9

The relevant sentence was removed as a result of simplifying the introduction following comments from other referees (L.106-107). However, in other places, as you suggested, we added authority or authority/family/order when the scientific name first appears. (L.79-80, 103-104, 300-302, 439).

>> L68-71: Maintain clarity: Recently in Japan, native pollinators like ABC and XYZ have been used to replace or supplement Apis mellifera in greenhouse strawberry cultivation. However, these native pollinators have not yet been consistently adopted across a large scale.<<

REPLY 10

As we responded in the REPLY 9, The relevant sentence was removed as a result of simplifying the introduction following comments from other referees. (L.105-108).

>> L71, 77 and 80: I suggest using the first experiment, second experiment, and third experiment. <<

REPLY 11

We followed your comment, like REPLY 7. (L.93, 110, 117, 120).

>> Figure 1b: I would label the figure 1b for educating the early career researchers. And describe it a little more in the caption. <<

REPLY 12

Following your comment, we labeled the figure 1b and added the legend. (L.138-142).

>> L253: Please clarity whether it was ant nesting or ant infestation. Both are different.<<

REPLY 13

Sometimes ants nested inside the shells, and sometimes they infested the shells nested by O. orientalis. We added information about how data is handled when ants infested. (L.343-345).

>> L265: Remove see results for details. It is understood. <<

>> L276 and 304: Same comment as for L265. <<

REPLY 14

Following your comment, we removed ‘see results for details from several figure legends’. (L.359-360, 373, 409).

>> L306-308: Please expand the discussion around your previous study. Highlight the deficiencies/future directions and how did you expand further in the current study. This will educate the early researchers in the scientific process. <<

REPLY 15

Following other referee’s comment, we moved this sentence to the beginning of 4.1.1. We also added the sentence that there was no significant difference in the nesting rates in E. amaliae shells between this study and our previous study. (L.414-420).

>> L311: What could be the reasons for higher preference for this shell? Is it the diameter or some architectural features? Think of the body diameter, larval size and other factors here. <<

REPLY 16

We have already written why small or large shells are not preferred, but we have dug a little deeper by adding new literatures. (L.422-442).

>> L327: How are these consistent with earlier study? Do explain when you say this.<<

REPLY 17

Following your comment, we have reworded the text to highlight how these results are consistent with our previous studies. (L.446-448).

Whenever we say ‘these results are consistent with our previous research’, we always added how it is consistent. (L.428-430).

>> L344: Please also discuss the possibility of using other biomaterials, as these plastic-based shells will add the pollution factor. So, if we can construct some shells using biomaterials and 3D printers, then what could be the possibilities and what could be the alternate materials which could be used. That may also provide the base for future studies.

Add one paragraph of future directions.<<

REPLY 18

Following your comment, we added that the use of normal plastics has a negative impact on the global environment, by citing literature. Next, we stated that a future challenge would be to use biodegradable biomass plastics as materials for artificial shells. (L.488-493).

>> L359, 362, 366: Please delete experiment 1, 2, and 3 in brackets. It is understood and clear.<<

REPLY 19

Following your comment, we deleted the words ‘experiment 1, 2, and 3’ from the conclusion. (L.498, 502, 506).

>> Please move supplementary figures as the main figures. These are needed in the main text.<<

REPLY 20

Following your comment, we move supplementary two figures as the main figures 3 and 5. (L.238-240, 281-283).

Round 2

Reviewer 3 Report

Comments and Suggestions for Authors

The authors have addressed all the comments.